# Correlation between Polymorphisms of Vitamin D Metabolism Genes and Perianal Disease in Crohn’s Disease

**DOI:** 10.3390/biomedicines12020320

**Published:** 2024-01-30

**Authors:** Jessica Cusato, Carla Cafasso, Miriam Antonucci, Alice Palermiti, Alessandra Manca, Gian Paolo Caviglia, Marta Vernero, Angelo Armandi, Giorgio Maria Saracco, Antonio D’Avolio, Davide Giuseppe Ribaldone

**Affiliations:** 1Department of Medical Sciences, Division of Gastroenterology, University of Torino, 10126 Turin, Italy; jessica.cusato@unito.it (J.C.); cafassocarla@gmail.com (C.C.); alice.palermiti@gmail.com (A.P.); alessandra.manca@unito.it (A.M.); gianpaolo.caviglia@unito.it (G.P.C.); angelo.armandi@unito.it (A.A.); giorgiomaria.saracco@unito.it (G.M.S.); antonio.davolio@unito.it (A.D.); 2SCDU Infectious Diseases, Amedeo di Savoia Hospital, ASL Città di Torino, 10149 Turin, Italy; miriam.antonucci20@gmail.com; 3Gastroenterology-U, “Città della Salute e della Scienza” Hospital, 10126 Turin, Italy; mvernero@cittadellasalute.to.it

**Keywords:** polymorphism, vitamin D, metabolism, genes, perianal, Crohn’s disease, SNP, genetic, fistulizing, fistula

## Abstract

Although the role of vitamin D (VD) in the pathogenesis and progression of Crohn’s disease (CD) is known, the association between single-nucleotide polymorphisms (SNPs) of genes linked to vitamin D pathway and CD risk is still under study. Furthermore, no significant association has been previously found between these SNPs and perianal CD (pCD), a severe phenotypic manifestation of CD that may present as perianal fistula, abscess, and recto-vaginal fistula. Among the mechanisms involved in its pathogenesis, local inflammation and intestinal microbiota alteration are recognized. VD seems to act on these elements. The aim of this study was to evaluate the presence of an association between SNPs of genes coding for enzymes, transporters, and receptors involved in the VD pathway and the occurrence of pCD. Blood samples of 206 patients with CD, including 34 with pCD, were analyzed for *VDR*, *CYP27B1*, *CYP24A1*, and *GC* genetic variants. *VDR Apal* Aa genotype and *VDR BsmI* Bb genotype resulted in an association with pCD (*p* = 0.01 and *p* = 0.02, respectively). Our study demonstrates for the first time the impact of the polymorphisms of genes associated with the VD pathway on the onset of pCD. Future multicenter studies are needed to confirm these data.

## 1. Introduction

Crohn’s disease (CD) is an immune-mediated, chronic, remitting, relapsing, and progressive intestinal pathology of unknown etiology, which can involve any part of the gastrointestinal system [1]. CD falls within the definition of inflammatory intestinal diseases (IBDs) together with ulcerative colitis (UC).

CD occurs more frequently in countries with high socioeconomic development, with a particularly high incidence and prevalence in Europe and North America [2]. In recent decades, the incidence is constantly increasing even in developing countries. This seems to suggest a fundamental role of environmental risk factors in the development of the pathology [3]. Within Western countries themselves there is a different incidence between urban and rural areas: the population of the former is in fact more subject to cases of IBDs and this is probably due to the differences in lifestyle, which determine greater exposure to risk factors for CD [4].

Perianal disease (pCD) is a severe phenotypic presentation of CD. It can present in the form of perianal fistula, rectovaginal fistula, perianal abscess, and anal stenosis. It can occur before, at the same time as, or after the diagnosis of luminal intestinal disease. Its presence at the diagnosis of CD is associated with a higher risk of symptom severity, need for advanced therapies, surgical resection, and hospitalization, with consequent reduction in quality of life [5]. Knowledge regarding the pathogenesis of pCD is not complete. Two mechanisms appear to play the main role: epithelial–mesenchymal transition and tissue remodeling enzymes. The hypothesis of the involvement of epithelial–mesenchymal transition in fistula formation is supported by the recognition of transition cells in the fistulous tracts of patients with CD, high levels of TGF β in the areas between transition cells and epithelial cells, and an increase in expression of TNF and its receptor in transitional cells [6].

Vitamin D3, also called cholecalciferol, is a fat-soluble vitamin available in the body, with 80% endogenous origin and exogenous origin in the remaining 20% [7]. Its activity regulates calcium homeostasis and bone metabolism, but also influences the immune, neurological, and cardiovascular systems. Vitamin D, in addition to regulating the absorption of intestinal calcium, plays a role in intestinal homeostasis: it repairs the integrity and function of the mucosal barrier, also influencing its permeability, regulates the composition of the microbiota, and exerts its anti-inflammatory and immunoregulatory activity at the local level [8]. Numerous studies agree on the hypothesis of a protective role of vitamin D in CD through the maintenance of the integrity of the mucosal barrier, regulation of the microbiome, anti-inflammatory, and immunomodulatory activity. These mechanisms act on the pathogenetic factors that contribute to the onset of CD [9].

The association between single-nucleotide polymorphisms (SNPs) of genes involved in vitamin D metabolism and CD is the subject of multiple studies. These mainly analyze polymorphisms of the vitamin D receptor (*VDR*) gene: *Apal* (rs7975232), *BsmI* (rs1544410), *Taql* (rs731236), *Fokl* (rs10735810), and *Cdx2* (rs11568820). The 2013 meta-analysis by Xue et al. suggests likely correlations: the predisposition to the onset of CD in carriers of the “a” allele of the *Apal* polymorphism and an increased risk of developing CD in Caucasian subject carriers of the tt genotype of *Taql* [10]. Furthermore, a 2018 study showed that TaqI tt genotype is correlated with a 3-fold increase in risk of developing intra-abdominal fistulizing CD, but is not correlated with the onset of perianal fistula [11].

To the best of current knowledge, there are no association studies in the literature between vitamin D and pCD that have demonstrated a significant association. However, the role of vitamin D in modulating the composition of the microbiota, stimulating the increase of non-pathogenic bacteria and reducing inflammation, both elements altered in the pathogenesis of pCD, is well known. [9,12]. The objective of this study is to evaluate the existence of an association between the SNPs of the genes encoding enzymes, transporters, and receptors involved in the process of activation, inactivation, and activity of vitamin D and the onset of pCD in patients with CD.

## 2. Materials and Methods

A genetic study was conducted on CD patients followed at the IBD clinic of the “A.O.U. Città della Salute e della Scienza di Torino”, Turin, Italy. All patients with CD diagnosis according to ECCO criteria [13] randomly collected in IBD-biobank from its establishment (October 2016) [14] to March 2022 were included in this study. Patients were divided into two categories:Presence of perianal disease: includes patients with CD with a history of perianal fistula, perianal abscess, or rectovaginal fistula.Absence of perianal disease: includes all patients with CD without a history of perianal fistula, perianal abscess, or rectovaginal fistula.

The following data were retrospectively collected for all subjects: date of birth, sex, family history of IBD, disease location, date of CD diagnosis, date of last follow-up visit, presence of pCD. Regarding disease localization, the Montreal classification was used (L1, the terminal ileum; L2, colon; L3, ileo-colon; and L4, upper gastrointestinal tract) [15]. For patients with pCD, the following data were also collected: history of perianal fistula, classification of the fistula according to Geldof [16], presence of abscess, recto-vaginal fistula, and stenosis. The medical treatments for pCD were also recorded: antibiotics, thiopurines, adalimumab, infliximab, ustekinumab, and vedolizumab. Finally, the surgical details for pCD were evaluated: number of perianal operations, seton placement, temporary stoma, and definitive stoma.

Whole blood samples from all patients collected in EDTA tubes were analyzed at the Pharmacology and Pharmacogenetics laboratory of the “Amedeo di Savoia” Hospital in Turin. Genomic DNA was extracted using a semi-automatic extractor (MagnaPure Compact, Roche, Monza, Italy) and analyzed with Taqman probes (Thermofisher, Waltham, MA, USA) via real-time polymerase chain reaction (PCR) (LightCycler 480, Roche, Monza, Italy). The distribution of the genotypes of the polymorphisms of the genes reported in Table 1 was evaluated. 

The study followed the principles of the Declaration of Helsinki and was approved by the local ethical committee (Comitato Etico Interaziendale A.O.U. Città della Salute e della Scienza di Torino—A.O. Ordine Mauriziano—A.S.L. Città di Torino) (approval code 0109499).


*Statistical Analysis*


The statistical analysis was carried out through the following steps:Evaluation of the distribution of the genotypes in the two categories, perianal disease yes and perianal disease no.Calculation of the significance of the association between the two variables, genotype and pCD, with the chi-squared test and the calculation of the *p* value. The association between genotype and pCD was considered significant in case of *p* value < 0.05.In case of *p* value < 0.05, a second association study was carried out between pCD and genotype in which the categorical variable of the genotype includes two classes: heterozygous genotype and both the wild type and mutated genotype homozygosity.Calculation of the significance of the association between pCD and the presence of the heterozygous genotype with the chi-squared test and calculation of the *p* value. The association was considered significant in case of *p* value < 0.05.Logistic regression was performed by calculating the odds ratio (OR) and the 95% confidence intervals (CIs).

All the data described above were collected in an Excel database and analyzed using MedCalc^©^ software version 22.005-2023. Categorical variables were reported as number and percentages and analyzed with the chi-squared test. Multivariate analysis (logistic regression) was then carried out by entering the variables of clinical interest. CIs were calculated at 95% and statistical significance was defined at *p* values less than 0.05. In the presence of a statistically significant result, the OR was calculated.

## 3. Results

In total, 206 patients affected by CD were evaluated with mean follow up from CD diagnosis of 15 (±11) years.

### 3.1. Baseline Characteristics

Characteristics of the study population at the last follow-up visit are shown in Table 2. 

### 3.2. Characteristics of Patients with pCD

Characteristics of perianal disease in the 34 patients with pCD are reported in Table 3.

Of the 34 patients with pCD, 29 had perianal fistula: Geldof classification is reported in Table 4.

History of medical and surgical therapies for pCD is reported in Table 5.

### 3.3. Association between Vitamin D Pathway Gene Polymorphisms and the Presence of pCD

Frequency of vitamin D pathway gene polymorphisms in patients with and without pCD is reported in Table 6.

#### 3.3.1. *VDR BsmI* (rs1544410)

Out of the 34 patients with pCD, 8 (23.5%) presented the wild-type BB genotype, 22 (64.7%) the heterozygous Bb genotype, and 4 (11.8%) the homozygous mutated bb genotype. A comparison was made between those who present the Bb heterozygous genotype compared to those who, instead, present the wild type and homozygous allelic versions: BB + bb. Out of the total of 34 patients with pCD, 22 (64.7%) presented the heterozygous genotype, while 73/172 (42.4%) patients without pCD presented this allelic variant, *p* = 0.02, OR = 2.5 (95%CI 1.2–5.3) of having perianal disease, *p* = 0.02 (Figure 1).

#### 3.3.2. *VDR Apal* (rs7975232)

Out of the 34 patients with pCD, 3 (8.8%) presented the wild-type AA genotype, 25 (73.5%) the heterozygous Aa genotype, and 6 (17.5%) the homozygous mutated aa genotype. A comparison was made between those presenting the polymorphism with the Aa heterozygous genotype with those presenting the other two genotypes AA + aa. Out of the total of 34 patients with pCD, 25 (73.5%) presented the heterozygous genotype, while 84/172 (48.8%) patients without pCD presented this allelic variant, *p* = 0.009, OR = 2.91 (95%CI 1.3–6.6) of having perianal disease, *p* value = 0.01 (Figure 2).

## 4. Discussion

pCD is a severe phenotypic presentation of CD. It can present as perianal fistula, rectovaginal fistula, perianal abscess, and anal stenosis. The prevalence of pCD is around 20% among patients with CD [5], a prevalence similar to that of our study (17%).

Knowledge regarding the pathogenesis of pCD is not complete. Two mechanisms appear to play the main role: epithelial–mesenchymal transition and tissue remodeling enzymes [17,18]. In epithelial–mesenchymal transition, differentiated epithelial cells transform into mesenchymal cells and acquire the ability to migrate and penetrate adjacent tissues. This process is essential in embryogenesis and tumor and metastatic growth. Among the inducers of epithelial–mesenchymal transition, we recognize Transforming Growth Factor beta (TGFβ) and TNF produced by inflammatory cells, CD4 + and CD161 + helper T lymphocytes, CD20 + B lymphocytes, macrophages, neutrophils, and IL-13 produced by fibroblasts. Transformed cells express both epithelial cell markers, such as cytokeratin 8 and cytokeratin 20, and mesenchymal markers such as vimentin and smooth muscle actin. These cells reduce the expression of adhesion molecules such as E-cadherin and increase the transcription of factors such as SNAI1 and SLUG (or SNAI2). This leads to a loss of contact between cells, loss of apicobasal polarity and tissue remodeling. Furthermore, transition cells autonomously produce IL-13 and simultaneously express the IL-13 receptor, with autocrine signaling. This allows the transition cells to go deeper, a process facilitated by the increased expression of metalloproteases (MMPs), which degrade the components of the extracellular matrix with the formation of the fistulous tract. The hypothesis of the involvement of epithelial–mesenchymal transition and MMP in fistula formation is supported by the recognition of transition cells in the fistulous tracts of patients with CD, high levels of TGFβ in the areas between transition cells and epithelial cells, and an increase in expression of TNF and its receptor in transitional cells. Other molecules found to be mostly produced at the level of the fistulous tract are ETS1 (c-est-1 protein) and DKK1 (dickkopf-related protein 1). Epithelial–mesenchymal transition may also be involved in the pathogenesis of fistula-associated neoplasia. Increased MMP activity has been documented within fistulous tracts in patients with CD, particularly MMP3, MMP9, and MMP13 protein [6].

The intestinal microbiota appears to be involved in the pathogenesis of perianal fistulas as well. Different microorganisms were found in the study of fistulas from patients with pCD and idiopathic fistulas. In the former, the most frequent and numerous species are *Bradyrhizobium pachyrhizi*, followed by *Pseudomonas azotoformans*, and *Prevotella oris*. In the latter, however, the microbiota is homogeneous to the intestinal one. Further studies are necessary to confirm the role of the microbiota in the pathogenesis of pCD [19].

Further investigations were carried out on the genetic predisposition of CD and whether this may be involved in the pathogenesis of pCD. Two different studies analyzed the genetic factors associated with the development of perianal fistulas and intestinal fistulas and had different results [20,21]. The PUS10 gene, encoding pseudouridylate synthase 10, has a protective role against the development of pCD, and the C allele at the CDKAL1 variant (rs6908425) and the absence of NOD2 variants have been independently associated with perianal fistula. 

Currently, the association between vitamin D and CD is known. Vitamin D reduces the permeability of the intestinal mucosal barrier, modulates the microbiome by promoting the non-pathogenic bacteria, and reduces local inflammation, thus influencing the pathogenetic factors of CD [9]. 

Several studies have evaluated the influence of SNPs of the genes involved in the vitamin D pathway on the predisposition to the onset of CD. The 2013 meta-analysis by Xue et al. [10] suggests a probable correlation between SNPs of genes involved in vitamin D metabolism and the onset of CD: the predisposition to the onset of CD in carriers of the “a” allele of the Apal polymorphism and an increased risk of developing CD in European subject carriers of the tt genotype of TaqI. Both of these SNPs were evaluated in our study and we found that Apal with Aa genotype is significantly related to pCD compared to the other two genotypes (Aa vs. AA + aa, OR = 2.9, 95%CI = 1.3–6,.6, *p* = 0.01), suggesting that this genotype not only predisposes to CD in the general population, but also to a pCD phenotype in patients with CD. In our work, in addition to the Aa genotype of Apal, the Bb genotype of *BsmI* also appears to increase the risk of pCD in patients with CD when compared with the other two genotypes (Bb vs. BB + bb, OR = 2.5; 95%CI = 1.2–5.3, *p* = 0.02). The latter SNP was also studied by Dresner-Pollak et al. in 2004 and it was associated with increased susceptibility for UC in Ashkenazi Jewish patients [22]. *Apal* and *BsmI* are SNPs of the *VDR* gene located at the level of intron 8 in the 3’ region; therefore, they are genes that do not code proteins, and the study of their implication in various pathologies is therefore complex, as they do not modify the quantity or activity of the *VDR* receptor [23]. However, these polymorphisms could influence the gene expression by modifying the stability of *VDR* mRNA [24]. Another hypothesis is that these SNPs are “markers” for other functional polymorphisms of the *VDR* gene, or for other genes in linkage disequilibrium [25]. 

There are several studies analyzing the association between the heterozygous genotypes Aa of Apal and Bb of *BsmI* and an increase in indices of inflammation. In particular, dialysis patients carrying the recessive alleles “a” and “b” have a higher serum C-reactive protein CRP than carriers of the dominant alleles [26]. Furthermore, the “b” allele is correlated to a high CRP value even in cachectic cancer patients, a clinical characteristic that exposes these individuals to increased mortality [23]. In the literature, there is evidence of increased risk for some diseases based on the presence of these SNPs. In particular, Aa and Bb increase the risk of systemic lupus erythematosus [27] and multiple myeloma [28]. In addition, the heterozygous *BsmI* genotype increases the risk of mild cognitive impairment by 2-fold [29]. In the aforementioned studies, the heterozygous genotypes of Apal and *BsmI* are associated with the onset of inflammatory and tumor diseases and a worsening of the clinical outcome in various areas. This evidence supports the association evaluated in our study between these genotypes and pCD.

Among the SNPs analyzed, FokI and TaqI were not significantly related to pCD in our study. However, it has been demonstrated by other study groups that subjects carrying the FF genotype of FokI and subjects with the Tt + tt allelic variant of TaqI respond better to vitamin supplementation than those with other genotypes [30,31]. A 2018 study [11] demonstrated that TaqI tt genotype is correlated with a 3-fold increase in the risk of developing intra-abdominal fistulizing CD, but it is not correlated with the onset of perianal fistula, in agreement with our study. 

Our study has limitations: we analyzed biological samples from patients from only one hospital, a reference center for IBD, and therefore involved a reduced number of samples compared to other multicenter genetic studies with higher statistical power. Furthermore, the most frequent polymorphisms in the Caucasian population were analyzed and the concentration of vitamin D was not evaluated.

## 5. Conclusions

Our study demonstrates that the presence of the heterozygous genotype of Apal and *BsmI* significantly increases the risk of having pCD. A confirmation of these promising results with multicenter and multiethnic studies will be necessary.

## Figures and Tables

**Figure 1 biomedicines-12-00320-f001:**
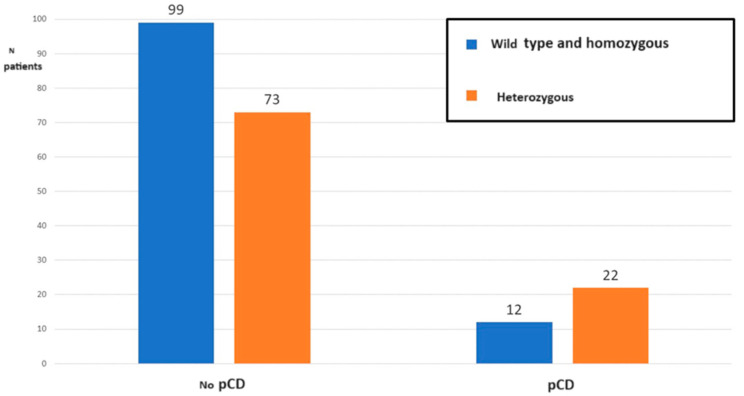
*BsmI* distribution (wild-type BB and homozygous mutated bb versus heterozygous Bb) in patients with Crohn’s disease with or without perianal Crohn’s disease.

**Figure 2 biomedicines-12-00320-f002:**
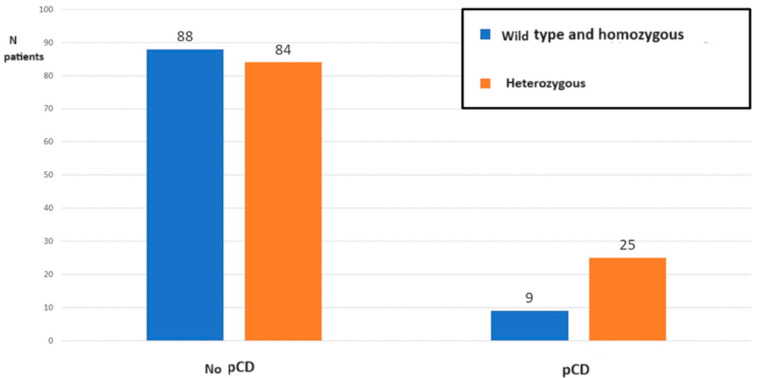
Apal distribution (wild-type AA and homozygous mutated aa versus heterozygous Aa) in patients with Crohn’s disease with or without perianal Crohn’s disease.

**Table 1 biomedicines-12-00320-t001:** SNPs of genes involved in vitamin D metabolism.

Gene	SNPs
** *VDR* **	*Apal* (rs7975232) C > A
	*BsmI* (rs1544410) G > A
	*Taql* (rs731236) T > C
	*Fokl* (rs10735810) T > C
	*Cdx2* (rs11568820) A > G
** *CYP27B1* **	*CYP27B1* +2838 (rs4646536) C > T
** *CYP24A1* **	*CYP24A1* 8620 (rs2585428) A > G
	*CYP24A1* 22776 (rs927650) C > T
	*CYP24A1* 3999 (rs2248359) T > C
** *GC* **	*VDBP GC*1296 (rs7041) A > C

**Table 2 biomedicines-12-00320-t002:** Characteristics of the whole population under study.

Variables, n/TOT (%)	
**Gender**	
Female	79/206 (38.3)
Male	127/206 (61.7)
**Family history of IBD**	
NO	175/206 (85.0)
YES	31/206 (15.0)
**Montreal Classification**	
L1	75/206 (36.4)
L1-L4	6/206 (2.9)
L2	21/206 (10.2)
L2-L4	1/206 (0.5)
L3	97/206 (47.1)
L3-L4	6/206 (47.1)
**Perianal Disease ***	
NO	172/206 (83.5)
YES	34/206 (16.5)

n = number; TOT = total; * perianal fistula, perianal abscess, or recto-vaginal fistula.

**Table 3 biomedicines-12-00320-t003:** Characteristics of perianal disease.

Characteristics, n/TOT (%)	
**Perianal fistula**	
NO	5/34 (14.7)
YES	29/34 (85.3)
**Abscess**	
NO	18/34 (52.9)
YES	16/34 (47.1)
**Recto-vaginal fistula**	
NO	28/34 (82.4)
YES	6/34 (17.6)
**Stenosis**	
NO	31/34 (91.2)
YES	3/34 (8.8)

n = number; TOT = total.

**Table 4 biomedicines-12-00320-t004:** Distribution of patients with perianal fistula according to Geldof classification.

Geldof Classification, n/TOT (%)	
**CLASS 1**	16/29 (55.2)
**CLASS 2nd**	7/29 (24.1)
**CLASS 2b**	5/29 (17.2)
**CLASS 2c-i**	0
**CLASS 2c-ii**	0
**CLASS 3**	0
**4th CLASS**	0
**CLASS 4b**	1/29 (3.4)

n = number; TOT = total.

**Table 5 biomedicines-12-00320-t005:** History of medical and surgical therapy in patients with pCD.

Therapy, n/TOT (%)	
**Antibiotics**	
FORMER	3/34 (8.8)
NO	14/34 (41.2)
CURRENT	17/34 (50.0)
**Thiopurines**	
FORMER	13/34 (38.2)
NO	18/34 (52.9)
CURRENT	3/34 (8.8)
**Adalimumab**	
FORMER	13/34 (38.2)
NO	7/34 (20.6)
CURRENT	14/24 (41.2)
**Infliximab**	
FORMER	6/34 (17.4)
NO	26/34 (76.5)
CURRENT	2/34 (5.9)
**Ustekinumab**	
FORMER	2/34 (5.9)
NO	26/34 (76.5)
CURRENT	6/34 (17.6)
**Vedolizumab**	
FORMER	2/34 (5.9)
NO	25/34 (73.5)
CURRENT	7/34 (20.6)
**Placement of setons**	
NO	22/34 (64.7)
YES	12/34 (35.3)
**Temporary stoma**	
NO	34/34 (100)
YES	0
**Definitive stoma**	
NO	32/34 (94.1)
YES	2/34 (5.9)

n = number; TOT = total.

**Table 6 biomedicines-12-00320-t006:** Association between vitamin D pathway gene polymorphisms and the presence of pCD.

Vitamin D Pathway Gene Polymorphisms	pCD w/He/Ho n/Tot (%)	No pCD w/He/Ho n/Tot (%)	*p* Value
***VDR Cdx2* (rs11568820)**	20/34 (58.8); 8/34 (23.5); 6/34 (17.7)	84/172 (48.8); 62/172 (36.1); 26/172 (15.1)	0.45
***VDR Fokl* (rs10735810)**	4/34 (11.8); 17/34 (50); 13/34 (38.2)	15/172 (8.7); 81/172 (47.1); 76/172 (44.2)	0.75
***VDR Taql* (rs731236)**	6/34 (17.7); 23/34 (67.6); 5/34 (14.7)	55/172 (32.0); 83/172 (48.3); 34/172 (19.8)	0.14
***VDR BsmI* (rs1544410)**	8/34 (23.5); 22/34 (64.7); 4/34 (11.8)	70/172 (40.7); 73/172 (42.4); 29/172 (16.9)	0.047
***VDR Apal* (rs7975232)**	3/34 (8.8); 25/34 (73.5); 6/34 (17.7)	32/172 (18.6); 84/172 (48.8); 56/172 (32.6)	0.025
***CYP24A1 22776* (rs927650)**	8/34 (23.5); 20/34 (58.8); 6/34 (17.7)	47/172 (27.3); 97/172 (56.4); 28/172 (16.3)	0.92
***CYP24A1 8620* (rs2585428)**	11/34 (32.4); 16/34 (47.1); 7/34 (20.5)	45/172 (26.2); 85/172 (49.4); 42/172 (24.4)	0.73
***CYP24A1 3999* (rs2248359)**	5/34 (14.7); 20/34 (58.8); 9/34 (26.5)	30/172 (17.4); 89/172 (51.7); 53/172 (30.8)	0.80
***GC 1296* (rs7041)**	6/34 (17.6); 21/34 (61.8); 7/34 (20.6)	49/172 (28.5); 80/172 (46.5); 43/172 (25.0)	0.29

pCD = perianal Crohn’s disease; w = wild type; He = heterozygous; Ho = homozygous mutated.

## Data Availability

Data supporting reported results are available upon request.

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
