# Peer review of "Correlation between Polymorphisms of Vitamin D Metabolism Genes and Perianal Disease in Crohn’s Disease"

_biomedicines, 2024, doi:10.3390/biomedicines12020320_

Round 1

Reviewer 1 Report

Comments and Suggestions for Authors

Limited evidence from preclinical and clinical studies suggests that the initiation, progression and maintenance of CD-associated perinial fistulas involves complex interactions between host, microbial and environmental factors.

Authors are suggested to add more references and current publications in the field.

Comments on the Quality of English Language

English language may be improved but acceptable in the present form. It is interesting and easy to understand. 

Author Response

Q1.     Limited evidence from preclinical and clinical studies suggests that the initiation, progression and maintenance of CD-associated perinial fistulas involves complex interactions between host, microbial and environmental factors.

Authors are suggested to add more references and current publications in the field.

A1. Dear Reviewer, thank you very much for your suggestion. We added more references and current publications in the field about CD-associated perianal fistulas pathogenesis.

Reviewer 2 Report

Comments and Suggestions for Authors

The article presented by Jessica Cusato and collaborators, entitled “Correlation between polymorphisms of vitamin D metabolism  genes and perianal disease in Crohn's disease”, is an original study that aimed to analyze the association between single-nucleotide polymorphisms of  genes coding for enzymes, transporters, and receptors involved in the Vitamin D pathway and the occurrence of perineal Crohn’s Disease. To develop the objective, they analyze the genomic DNA (obtained from blood) of 206 Crohn's patients using real-time polymerase chain reaction. Of the 206 patients, 34 were diagnosed with perianal disease, which represents 16.5% of their cohort and is therefore within what has been described in other works. The contribution of the manuscript to scientific literature is medium-low. The work is mainly descriptive without delving into possible mechanisms and with the absence of in vitro work.

The work is well written and is well understood, the figures and tables are correct. The objective is clear and the execution is correct. I only have minor comments

Minor revision:

1.      Line 19. Acronym SPNs

2.      Line 26. The authors must specify the Biological samples used (biopsies? blood?).

3.      Line 52. Knowledge regarding the pathogenesis of pCD is not complete. Two mechanisms appear to play the main role: epithelial-mesenchymal transition (EMT) and tissue remodeling enzymes. The authors must provide more scientific evidence (references) about the two mechanisms.

4.      Line 87. The authors must detail how much time they needed to obtain the cohort of 206 patients (the start and end yea), in addition to indicating the number of patients in this section.

5.      Figures should show a more detailed description in the figure caption. The texts inside the figure begin with Capital Letters

Author Response

The work is well written and is well understood, the figures and tables are correct. The objective is clear and the execution is correct. I only have minor comments.

Dear reviewer, thank you for appreciating our manuscript. We have corrected it according to your suggestions and hope that it is now suitable for publication.

Q1. Line 19. Acronym SPNs.

A1. We added the definition.

Q2. We specified “blood”.

A2. Thank you for your suggestion. We added a paragraph at the beginning of the introduction to introduce what IBDs are and what unmet needs are.

Q3. Line 52. Knowledge regarding the pathogenesis of pCD is not complete. Two mechanisms appear to play the main role: epithelial-mesenchymal transition (EMT) and tissue remodeling enzymes. The authors must provide more scientific evidence (references) about the two mechanisms.

A3. We added more scientific evidence (references) about the two mechanisms in the Discussion section.

Q4. Line 87. The authors must detail how much time they needed to obtain the cohort of 206 patients (the start and end yea), in addition to indicating the number of patients in this section.

A4. We specified “from October 2016 to March 2022".

Q5. Figures should show a more detailed description in the figure caption. The texts inside the figure begin with Capital Letters

A5. We have corrected the captions and figures accordingly.